# Understanding coordination reaction for producing stable electrode with various low work functions

Hirohiko Fukagawa [1✉], Kazuma Suzuki[2], Hirokazu Ito[2], Kaito Inagaki[2], Tsubasa Sasaki[1], Taku Oono[1], Munehiro Hasegawa[3], Katsuyuki Morii[3,4] & Takahisa Shimizu [1]

The realisation of a cathode with various work functions (WFs) is required to maximise the potential of organic semiconductors that have various electron affinities. However, the barrier-free contact for electrons could only be achieved by using reactive materials, which significantly reduce the environmental stability of organic devices. We show that a stable electrode with various WFs can be produced by utilising the coordination reaction between several phenanthroline derivatives and the electrode. Although the low WF of the electrode realised by using reactive materials is specific to the material, the WF of the phenanthroline-modified electrode is tunable depending on the amount of electron transfer associated with the coordination reaction. A phenanthroline-modified electrode that has a higher electron injection efficiency than lithium fluoride has been demonstrated. The observation of various WFs induced by the coordination reaction affords strategic perspectives on the development of stable cathodes unique to organic electronics.

[1] Japan Broadcasting Corporation (NHK), Science & Technology Research Laboratories, 1-10-11 Kinuta, Setagaya-ku, Tokyo 157-8510, Japan. [2] Department of Physics, Graduate School of Science, Tokyo University of Science, 1–3 Kagurazaka, Tokyo 162-8610, Japan. [3] Nippon Shokubai Co., Ltd., 5-8 Nishi Otabi-Cho, Suita, Osaka 564-8512, Japan. [4] Nippon Shokubai Research Alliance Laboratories, Osaka University, 2-1 Yamadaoka, Suita, Osaka 565-0871, Japan. ✉email: fukagawa.h-fe@nhk.or.jp

O rganic electronics such as organic light-emitting diodes (OLEDs), organic thin-film transistors (OTFTs), organic solar cells (OSCs), perovskite solar cells (PSC) and organic semiconductor laser diodes (OSLDs) have made significant progress in recent years. In addition, there have been many reports on flexible optoelectronic devices such as displays, lighting and sensor arrays, which have many advantages such as versatility in shape and light weight[1–6]. Since barrier-free contacts for holes and electrons are essential for efficient organic optoelectronic devices, an anode with a high work function (WF) and a cathode with a low WF are required. Surface WFs of anodes have been tuned to over 5 eV by utilising p-type dopants that have large electron affinity (EA) and/or metal oxides such as molybdenum trioxide[7–10]. On the other hand, the electron injection/collection efficiency at cathode/ organic semiconductor interfaces has mainly been tuned by using reactive materials such as alkali metals that have a low WF below 3 eV[10–13]. However, the applicability of alkali metals is proved to be limited by their high reactivity and diffusivity[14]. In particular, flexible optoelectronic devices using reactive materials require stringent encapsulation, which hinders the widespread proliferation of flexible devices.

The method of improving the electron injection/collection efficiency without using reactive materials has been the subject of intense study in recent years[15–22]. As a replacement for alkali metals, many molecular n-dopants with low ionisation potential (IP) have been evaluated. The electron-injection barrier can be tuned by the electron transfer (ET) from n-dopants to organic semiconductors. The development of stable n-dopants has been hindered by the energy-level matching between the organic semiconductor and the dopant, since the stability is compromised by the lower IP. On the other hand, improvement of the electron injection/collection efficiency utilising the electron exchange between the nitrogen atom of organic compounds and inorganic materials has also been intensively studied[23–30]. Promising organic compounds include amine-containing molecules such as polyethyleneimine (PEI), which can tune the surface WF of about 1 eV by utilising both the ET from PEI to the electrode surface and the orientation of the molecular dipole of PEI[23]. Other promising organic compounds are phenanthroline (Phen) derivatives that form a coordination bond with metals such as Ag[25–30]. Yoshida[26] clarified that the high electron injection/collection efficiency at the Ag electrode/bathocuproine (BCP) interface is caused by the formation of a Ag-BCP complex. Bin et al.[30] reported that the electron-injection efficiency of a Ag-doped 4,7-dimethoxy-1,10-phenanthroline (p-MeO–Phen) thin film is higher than not only that of a Ag-doped BPhen thin film, but also that of Cs-doped p-MeO–Phen. This higher electron-injection efficiency is achieved by the larger electron exchange between p-MeO–Phen and Ag than that between BCP and Ag, which is caused by introducing the electron-donating dimethoxy substituent into Phen. The WF of the Ag cathode can be largely decreased from 4.2 to 2.8 eV when Ag-doped p-MeO–Phen is deposited. Thus, there is a possibility that the WF of an electrode, which has a specific value for the electrode material, can be tuned freely by utilising the interaction between metals and Phen derivatives with different electron-donating moieties. High tunability of the WF is desired, especially for the cathode, since the EA of organic semiconductors strongly depends on the materials used in devices: the EA of organic semiconductors suitable for OSCs, OTFTs and PSCs is about 4 eV, whereas that of organic semiconductors suitable for OLEDs is less than 3 eV[11,15]. The realisation of a barrier-free contact for electrons without using reactive materials and/or conventional n-dopants will also accelerate the practical application of flexible optoelectronic devices[5,6].

Here, we report a universal method of producing a stable electrode with various WFs by utilising the coordination reaction between five Phen derivatives and the electrode. The surface WF of the electrode can be tuned in the range of 3.29–2.43 eV by forming thin films of a Phen derivative on ITO/ZnO, although the WF of the ITO/ZnO electrode is 4.10 eV. It was found from density functional theory (DFT) calculation that the change in the WF ($\Delta_{\mathrm{WF}}$) has a strong correlation with the amount of ET associated with the coordination reaction. The electron-injection efficiency of the electrode basically corresponds to its WF. Although the electron-injection efficiency of the electrode modified with p-MeO–Phen is lower than that of lithium fluoride (LiF), the Phen-modified electrode having the lowest WF shows higher electron-injection efficiency than LiF. Furthermore, the air stability of the Phen-modified electrode is about ten times higher than that of LiF, and the operational stability of the OLED using the Phen-modified electrode is comparable to that of the OLED using LiF.

## Results

**Effect of coordination reactions on ΔWF.** The effect of the coordination reaction of Phen derivatives on the change in the surface WF of electrodes was investigated by evaporating five Phen derivatives on ITO/ZnO. There are three reasons why we selected ITO/ZnO as a stable metal-oxide electrode. Firstly, the WF of ITO is sensitive to its surface condition[31]. Secondly, the WF of ZnO is smaller than that of ITO; thus, ITO/ZnO is more favourable for both producing a low WF and enhancing the electron injection from ITO[5,32]. Since the surface WF is determined by the average of the point charges on the surface, it is reasonable to suppose that the reductions of both the surface WF observed by ultraviolet photoemission spectroscopy (UPS) measurements and the electron-injection barrier are independent of the growth behaviour of ZnO except in the case of large-island growth under special growth conditions[33,34]. Lastly, the changes in the surface WF observed by the deposition of Phen can be discussed as the effect of the coordination reaction, since there is no push-back effect on metal-oxide surfaces unlike metal surfaces[35]. The chemical structures of the Phen derivatives used in this study are shown in Fig. 1a. It has been proposed that a molecular design to increase the nucleophilicity around two N atoms in the Phen unit, which are considered as the main binding sites with a metal atom, is essential to reduce the surface WF of Phen–metal complexes (Fig. 1b)[30]. The molecular electrostatic potential (ESP) of Phen derivatives, which indicates the nucleophilicity, was calculated in a previous study, and p-MeO–Phen was found to have a stronger nucleophilicity than 4,7-diphenyl-1,10-phenanthroline (BPhen)[30]. In accordance with this molecular design strategy, we introduced electron-donating groups to increase the nucleophilicity of the Phen derivatives. The ESP of several Phen derivatives was determined by DFT calculation. As a result, two Phen derivatives, 4,7-Bis(dimethylamino)-1,10-phenanthroline (p-NMe$_2$-Phen) and 4,7-bis(1-Pyrrolidinyl)-1,10-phenanthroline (p-Pyrrd-Phen), were found to have a larger ESP than p-MeO–Phen (Supplementary Fig. 1). In addition to the above-mentioned four Phen derivatives, 4,7-di(9H-carbazol-9-yl)-1,10-phenanthroline (BUPH1)[36], the ESP of which is smaller than that of BPhen, was used for comparison. Although there have been many reports on the electronic structure and the related interactions between a specific Phen derivative and several metals, there have been few reports on the interactions between several Phen derivatives and a specific metal[26,37,38]. Bin et al.[30] were the first to use three Phen derivatives, and the p-MeO–Phen–Ag complex has been reported to be an excellent electron-injection layer (EIL). However, the three types of Phen

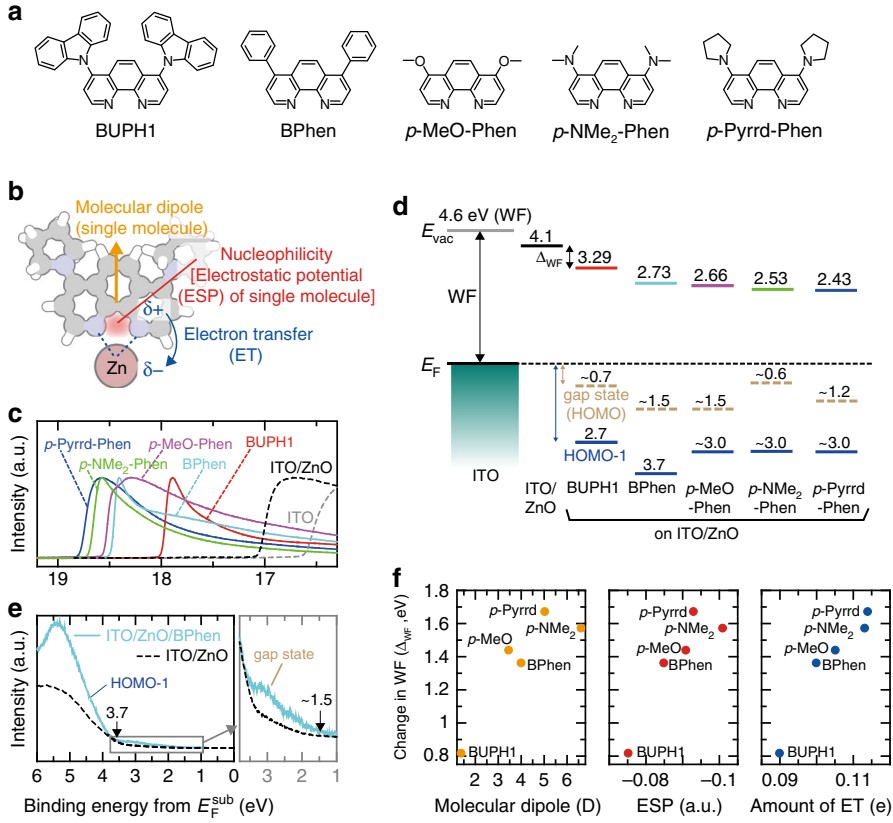

**Fig. 1 Change in work function caused by coordination reaction. a** Chemical structure of the Phen derivatives used in this study. **b** Schematic illustration of coordination reaction between *p*-Pyrrd-Phen and Zn. **c** HeI UPS spectra of electrode and Phen derivatives (5 nm) on ITO/ZnO in the secondary region. **d** Energy-level diagram of Phen derivatives on ITO/ZnO estimated from UPS results. **e** HeI UPS spectra of 5-nm-thick BPhen on ITO/ZnO in the valence band region. **f** Summary of the correlation between $\Delta_{WF}$ (vertical axis) and three calculated parameters (horizontal axis) that likely correlate with $\Delta_{WF}$.

used here do not provide a systematic understanding of the correlation between coordination reactions and $\Delta_{WF}$ around the cathode. We see from the calculated ESP that both *p*-NMe₂–Phen and *p*-Pyrrd–Phen have the potential to be better EILs than *p*-MeO–Phen when they are used around the cathode. The study of the interactions of five different Phen derivatives with metals will provide a systematic understanding of the effect of coordination reactions on $\Delta_{WF}$.

Figure 1c shows the results of UPS measurements on a series of Phen derivatives deposited on an ITO/ZnO substrate; the spectra revealed reductions in the WF from 4.10 to 3.29 eV for BUPH1, from 4.10 to 2.73 eV for BPhen, from 4.10 to 2.66 eV for *p*-MeO–Phen, from 4.10 to 2.53 eV for *p*-NMe₂–Phen, and from 4.10 to 2.43 eV for *p*-Pyrrd–Phen, as summarised in Fig. 1d. Before we discuss the observed $\Delta_{WF}$, it would be useful to see the UPS spectra for the valence band region shown in Fig. 1e. In the UPS spectrum of BPhen on ITO/ZnO, the gap state is observed near the Fermi level ($E_F$) as in the case with BCP–metal complex systems[26,37]. Such a gap state is also observed in other Phen derivatives on ITO/ZnO (Supplementary Fig. 2). Yoshida[26] reported on complex formation between BCP and a Ag atom by comparing the electronic structure obtained by UPS with the calculated orbital energy. Here, the molecular orbitals and orbital energies were calculated by placing a Zn atom near N in Phen as in a previous report (Supplementary Fig. 3)[26]. The highest occupied molecular orbital (HOMO) of the Phen derivative–Zn complex, which is the gap state, is localised on the Zn atom. On the other hand, the HOMO-1 of the Phen derivative–Zn complex mainly consists of the HOMO of each Phen derivative (Supplementary Figs. 1 and 3). For most Phen–Zn complexes,

the IPs obtained by UPS are highly correlated with the calculated IPs, suggesting complex formation between Phen and the Zn atom (Supplementary Fig. 4). Furthermore, the chemical interaction between *p*-Pyrrd–Phen and Zn was confirmed by X-ray photoelectron spectroscopy (Supplementary Fig. 5). Since the purpose of this study is to understand the effect of the coordination reaction of Phen derivatives on tuning the surface WF of electrodes, we analysed the observed $\Delta_{WF}$ using three calculated parameters that likely correlate with $\Delta_{WF}$, as illustrated in Fig. 1b. The intrinsic molecular dipole moment was estimated by DFT calculation with a single molecule as the ESP (Supplementary Fig. 1). In addition, it has been proposed that the formation of an interface dipole, which is caused by the slight ET associated with the coordination, can affect $\Delta_{WF}$ (Supplementary Fig. 3)[23]. Before we discuss the correlation between the ET from Phen to the metal and $\Delta_{WF}$, we must clarify the validity of using the amount of ET calculated by placing a Zn atom near N in Phen derivatives (Supplementary Fig. 3). We compared the amount of ET from ethylamine to Zn with a periodic ZnO structure reported by Zhou et al.[23] with the amounts of ET from ethylamine to metallic counterparts that we can treat in our calculations such as simple elements (Zn atom, ZnO) (Supplementary Fig. 6). We found that the Zn atom is more suitable than ZnO for the calculation of the ET from N to a metal without a periodic structure. The correlations between $\Delta_{WF}$ and three parameters are summarised in Fig. 1f. The fact that the amount of ET shows the strongest correlation with $\Delta_{WF}$ suggests that ET has the most effect on $\Delta_{WF}$. It was found that the calculated amount of ET of a two-molecule system could be a good indicator of $\Delta_{WF}$ caused by a coordination reaction.

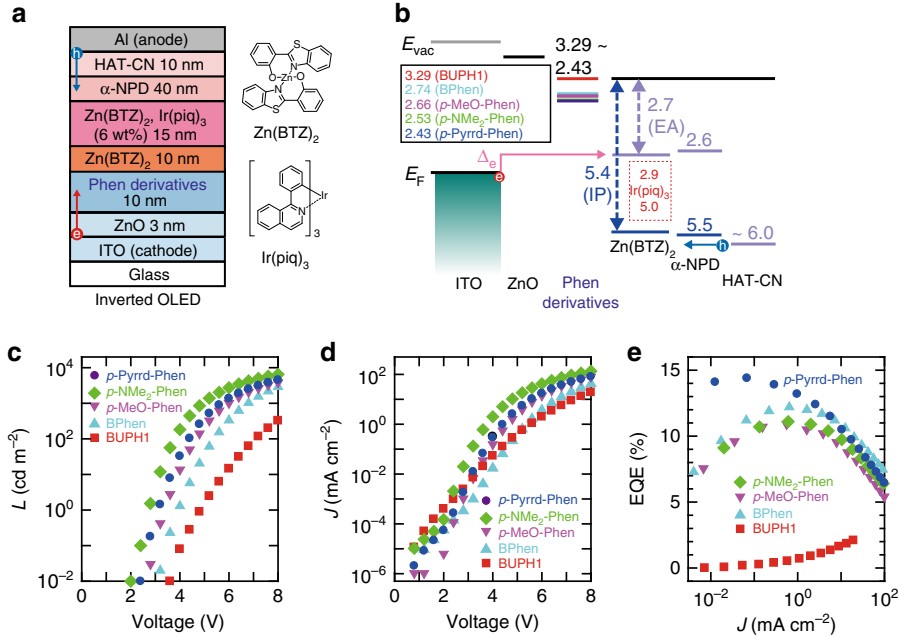

**Fig. 2 Schematic illustrations of inverted OLEDs and their characteristics. a** Multilayer structure of an inverted OLED and chemical structure of the materials used in the emitting layer. **b** Energy-level diagram of inverted OLEDs. **c** Luminance ($L$)–voltage, **d** current density ($J$)–voltage and **e** EQE–$J$ characteristics of inverted OLEDs prepared using five Phen derivatives.

**Fabrication and performance of inverted OLEDs.** The effect of the coordination reaction on the electron-injection/hole-blocking property was investigated by examining the Phen derivative-dependent characteristics of inverted OLEDs, as shown in Fig. 2a. Since the vacuum-level alignment model is assumed at organic/organic interfaces, the assumed energy-level matching in the inverted OLED using BUPH1 is illustrated in Fig. 2b[39]. The electron-injection barrier is indicated by $\Delta_e$, and it is expected that $\Delta_e$ decreases as $\Delta_{WF}$ increases upon changing the Phen derivative. We see from Fig. 2c, d that, basically, the larger the $\Delta_{WF}$, the lower the driving voltage. Although the driving voltages of both inverted OLEDs using $p$-NMe$_2$–Phen and $p$-Pyrrd–Phen are lower than that of the inverted OLED using $p$-MeO–Phen, the driving voltage of inverted OLEDs using $p$-NMe$_2$–Phen is the lowest, which is not consistent with $\Delta_{WF}$. We see from Fig. 2e that the external quantum efficiency (EQE) of the inverted OLED using $p$-MeO–Phen is lower than that of the inverted OLED using BPhen, although $p$-MeO–Phen causes a larger $\Delta_{WF}$ than BPhen. Thus, it is reasonable to assume that the characteristics of inverted OLEDs are dominated by not only $\Delta_{WF}$, but also the hole-blocking property around a cathode, as Bolink et al.[32] reported. This is because the holes are effectively injected or generated at 4,4′-bis[N-(1-naphthyl)-N-phenyl-amino]biphenyl (α-NPD)/1,4,5,8,9,11-hexaazatriphenylenehexacarbonitrile (HAT-CN)/Al interfaces[40]. The highest EQE of the inverted OLED fabricated using BPhen at a high current density is expected to originate from the strongest hole-blocking property at the BPhen/emitting layer interface. It is reasonable to suppose that the hole-blocking property at the Phen derivative/emitting layer interface is determined by the orbital energy of the HOMO of each Phen derivative (not the complex with Zn), whereas the hole-blocking property at the Phen derivative/ZnO interface is determined by the orbital energy of the gap state. We see from the difference in the energy of the gap state, as shown in Fig. 1d, that the hole-blocking property of the $p$-Pyrrd–Phen–Zn complex around the cathode is expected to be superior to that of the $p$-NMe$_2$–Phen–Zn complex, resulting in the higher EQE in the inverted OLED using $p$-Pyrrd–Phen. The fact that the

$p$-Pyrrd–Phen–Zn complex combines good electron-injection and hole-blocking properties suggests that $p$-Pyrrd–Phen is the most promising material for producing a Phen-modified electrode with a low WF. Since the holes that reach the Phen derivative make the interpretation of the characteristics of inverted OLEDs complicated, it is difficult to simply discuss the correlation between $\Delta_{WF}$ and the electron-injection efficiency in inverted OLEDs with ITO/ZnO as the cathode, which was used for the evaluation of $\Delta_{WF}$ in Fig. 1. However, the characteristics of inverted OLEDs shown in Fig. 2 greatly contributed to the evaluation of the air stability of the electrode modified by Phen derivatives, as will be shown in Fig. 3. The correlation between the value of $\Delta_{WF}$ derived from the Phen derivatives and the electron-injection efficiency, which is of great significance toward the realisation of organic devices without using reactive metals, was successfully clarified by using Phen derivatives for the EIL of conventional OLEDs (cOLEDs) as will be shown in Fig. 4.

The electron-injection efficiency and air stability of the low-WF electrode modified by $p$-Pyrrd–Phen were investigated by comparing the characteristics of four inverted OLEDs having various cathode-modification and electron transport layers (CMETLs) as shown in Fig. 3a. The configuration of the four inverted OLEDs is basically the same as that of the inverted OLED in Fig. 2a, except for the CMETL. We employed spin-coated $p$-Pyrrd–Phen in inverted OLED-2 since the solution processing promotes the penetration of ZnO into the $p$-Pyrrd–Phen layer, which significantly improves electron injection in inverted OLEDs[5]. In addition, inverted OLEDs using a combination of ZnO and PEI, which is the combination most widely used as an EIL[23], and inverted OLEDs using LiF, which is a widely used EIL in conventional OLEDs, are denoted as inverted OLED-3 and inverted OLED-4, respectively. We see from Fig. 3b, c that the driving voltage of inverted OLED-2 is similar to that of inverted OLED-3, and the driving voltage of these two inverted OLEDs is much lower than that of inverted OLED-1. The interpenetrated ZnO in inverted OLED-2 may contribute not only directly to electron injection/transport, but also to an increase in the number of coordination reactions

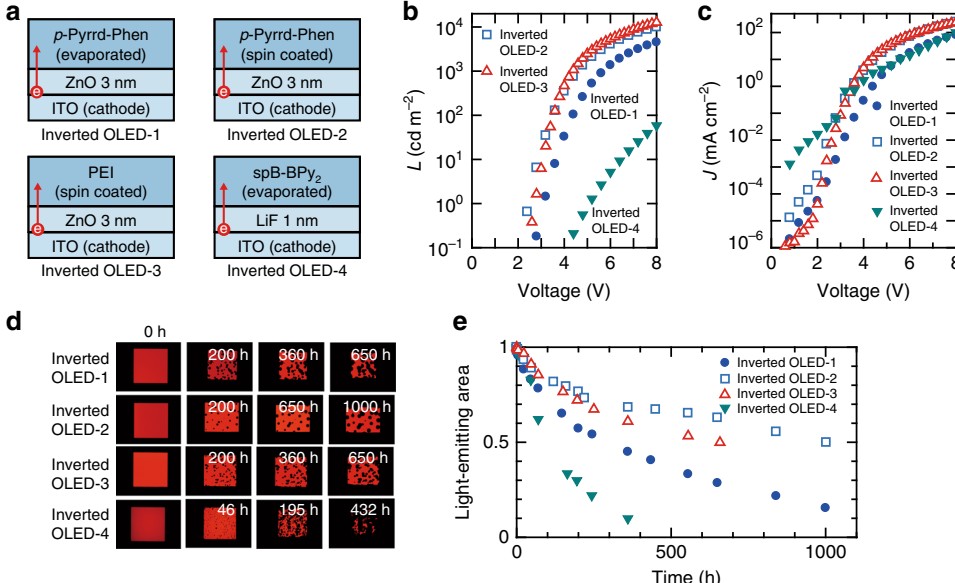

**Fig. 3 Evaluation of CMETLs in inverted OLEDs. a** Schematic illustration of CMETLs. **b** Luminance (*L*)–voltage and **c** current density (*J*)–voltage characteristics of inverted OLEDs prepared using four CMETLs. **d** Images of light-emitting areas of inverted OLEDs without encapsulation as a function of storage time. Although dc current was applied to the inverted OLEDs at the time of measurement, dc current was not applied under the storage condition. **e** Decay in light-emitting area of inverted OLEDs.

between *p*-Pyrrd–Phen and Zn, resulting in a lower driving voltage than that of inverted OLED-1. The driving voltages of both inverted OLED-1 and inverted OLED-2 are lower than that of inverted OLED-4. These results are reasonable since the WF of *p*-Pyrrd–Phen on Zn is smaller than those of both LiF and Li, which are 7 and 3 eV, respectively[41]. The low electron-injection efficiency in inverted OLED-4 may originate from the limited interaction between the initially deposited LiF and the subsequently deposited 6″-(4-([2,2′-bipyridin]-6-yl)-2-(5H-dibenzo[b, d]borolyl)phenyl)-2,2′:6′,3″-terpyridine (spB-BPy₂)[42]. That is because the chemical interaction between the electron-transporting material and LiF is essential to improve the electron-injection efficiency[43,44]. Actually, efficient electron injection is observed in the conventional OLED, where LiF is deposited on spB-BPy₂, as will be shown in Fig. 4. The air stability of each CMETL was evaluated by destroying the encapsulation glass of the fabricated inverted OLEDs. Figure 3d shows optical microscopy images of light-emitting areas of inverted OLEDs without encapsulation as a function of storage time. The decay in the light-emitting area is summarised in Fig. 3e, where the light-emitting area before exposure to the atmosphere is set to 1. The light-emitting area decreases in all inverted OLEDs after exposure to the atmosphere; however, the decay rate strongly depends on the CMETL. The results of our experiment clearly demonstrate that the air stability of the Phen-modified cathode is higher than that of the LiF-modified cathode. In particular, the half-life of the emitting area of inverted OLED-2 employing spin-coated *p*-Pyrrd–Phen is about ten times longer than that of the inverted OLED-4 employing LiF. Since the nonluminescent area increases with increasing the number of sites where electrons cannot be injected, the difference in the decay rate between inverted OLED-1 and inverted OLED-2 may originate from the difference in the number of coordination reactions, which is caused by the difference in the fabrication process. It is reasonable to suppose that the Phen-modified electrode with more electron injection sites is effective for reducing the decay rate. The air stabilities of inverted OLED-2 and inverted OLED-3, both of which have the spin-coated CMETL, are higher than that of inverted OLED-1. However, it is difficult to discuss the difference in decay rate in

detail since both the chemical structure and the film thickness of the CMETL are different. It is concluded from the results for the inverted OLEDs that the electron-injection efficiency and air stability of the Phen-modified electrode are higher than those of the LiF-modified electrode.

**Fabrication and performance of conventional OLEDs.** The correlation between the value of $\Delta_{WF}$ derived from the Phen derivatives and the electron-injection efficiency has successfully been observed by evaluating the EIL-dependent characteristics of a cOLED having the device configuration depicted in Fig. 4a, employing various EILs[45]. The emitting host we selected here is a thermally activated delayed fluorescent (TADF) material named 2,4-diphenyl-6-bis(12-phenylindolo)[2,3-a] carbazol-11-yl)-1,3,5-triazine (DIC-TRZ), which is ideal for demonstrating an efficient and operationally stable OLED[46]. Although some evaporated Al atoms diffuse into Phen derivatives in cOLEDs[47], we see from Fig. 4b, c that the electron-injection efficiency of the Phen-modified Al electrode basically corresponds to $\Delta_{WF}$ shown in Fig. 1d (Supplementary Fig. 7). The result of our experiment clearly shows that the electron-injection efficiency can easily be controlled by changing the Phen derivative around the cathode. Furthermore, we show that a Phen derivative can be a feasible alternative to popular but reactive EILs such as LiF and 8-quinolinolato lithium (Liq)[48]. To discuss the difference between this study and the previous study in detail, the characteristics of cOLEDs employing *p*-Pyrrd–Phen, *p*-OMe–Phen, LiF and Liq are summarised in Fig. 4d–g. We see from Fig. 4d, e that the driving voltages of the two cOLEDs employing *p*-Pyrrd–Phen and *p*-MeO–Phen are lower than that of the cOLED without an EIL, and the cOLED employing *p*-Pyrrd–Phen exhibits a much lower driving voltage. Furthermore, the electron-injection efficiency of the *p*-Pyrrd–Phen-modified Al electrode is demonstrated to be higher than that of both LiF and Liq-modified Al electrodes from the fact that the driving voltage of the cOLED employing *p*-Pyrrd–Phen is the lowest. Since the configuration of the cOLED is optimised for the use of LiF or Liq, the EQE of the cOLED employing *p*-Pyrrd–Phen is slightly lower than those of the other

                    **5**

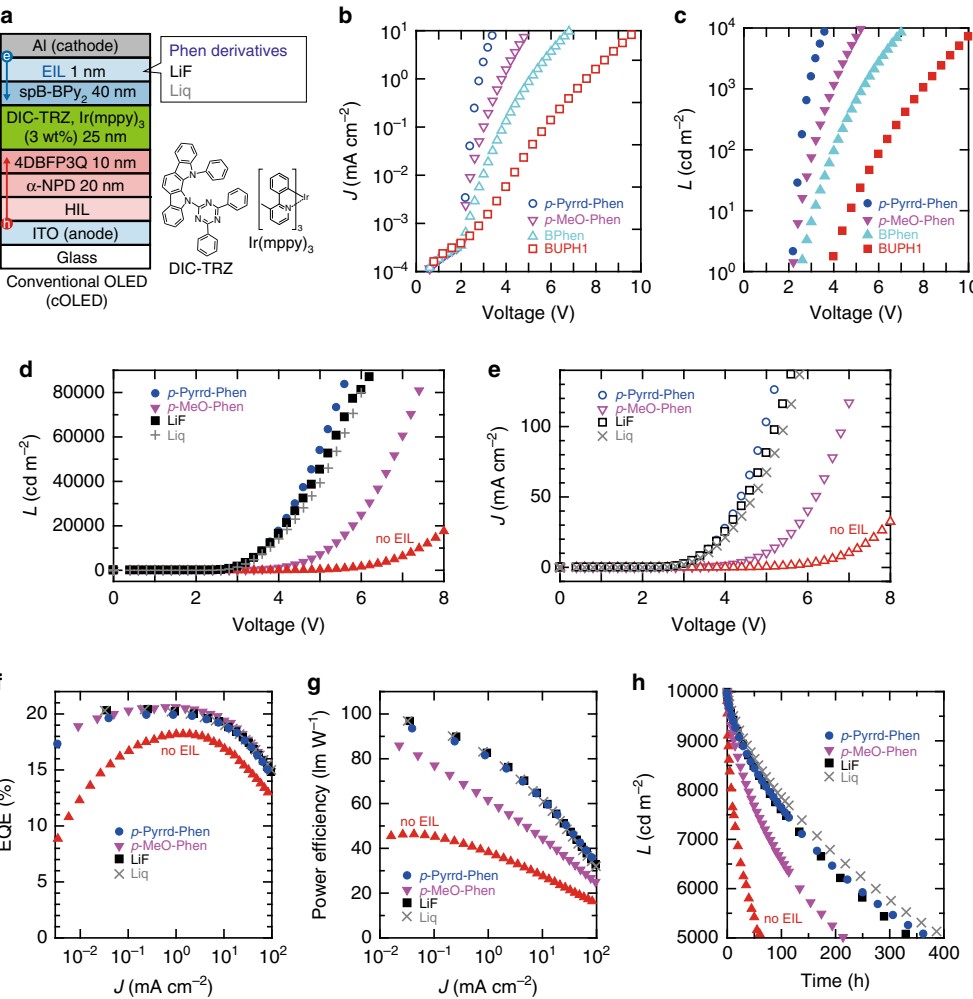

**Fig. 4 Evaluation of EIL in cOLEDs. a** Multilayer structure of a cOLED and chemical structure of the materials used in the emitting layer. **b** Luminance (L)–voltage and **c** current density (J)–voltage of cOLEDs prepared using four Phen derivatives. **d** L–voltage and **e** J–voltage characteristics of cOLEDs prepared using four EILs. **f** EQE–J curves of cOLEDs. **g** Power efficiency–J curves of cOLEDs. **h** Luminance–time characteristics of devices under a constant dc current.

cOLEDs as shown in Fig. 4f. Although the carrier balance in the cOLED employing *p*-Pyrrd–Phen is worse than those in the other two cOLEDs employing LiF and Liq, the operational stabilities of these three cOLEDs are comparable as shown in Fig. 4h. These results clearly indicate the high operational stability of *p*-Pyrrd–Phen. These high performances of the cOLED employing *p*-Pyrrd–Phen, such as the driving voltage, EQE, power efficiency and operational stability, are comparable to those of state-of-the-art green cOLEDs reported in the literature[49]. On the other hand, the relatively low operational stability of the cOLED employing *p*-MeO–Phen, where the electron-injection efficiency is insufficient, may be caused by the accumulation of the carrier recombination region rather than the stability of *p*-MeO–Phen[50–53]. The electron-injection efficiency of *p*-Pyrrd–Phen-modified Al is demonstrated to be higher than that of *p*-MeO–Phen-modified Al[30]. Although Ag-doped *p*-MeO–Phen may exhibit similar electron-injection efficiency to *p*-Pyrrd–Phen-modified Al, it is not easy to apply such a Ag-doped film to an OLED production line, where the organic layer deposition and metal deposition chambers are generally separated (Supplementary Table 1)[54]. It is no wonder that the EQEs of the OLEDs employing Phen derivatives are almost independent of the driving voltage since the EQEs independent of the driving

voltage have also been reported for another phosphorescent OLEDs fabricated using a TADF material as the host[55,56]. On the other hand, we succeeded in injecting electrons effectively from the *p*-Pyrrd–Phen-modified Al, and the difference in electron-injection efficiency between these two Phen derivatives can be seen from the difference in the coordination reaction. To the best of our knowledge, this is the first demonstration that an electrode modified by a strong coordination reaction can have a larger electron-injection efficiency than reactive materials. We would also like to mention an advantage of Phen derivatives with a high electron-injection efficiency such as *p*-Pyrrd–Phen over PEI, which is widely used as an EIL in inverted OLEDs[57,58]. Amine-based polymers such as PEI have also been used in OSCs as a cathode modification layer instead of BCP and/or BPhen owing to their high WF tunability[59]. However, there are two major problems for the practical application of PEI since PEI can only be prepared by a solution process. One is that it is not easy to prepare multilayer structures by a solution process when PEI is applied to devices with a top cathode such as cOLEDs[60]. The other is that the characteristics of devices fabricated using PEI are rather sensitive to the thickness of the PEI film, even though it is difficult to precisely control the film thickness by a solution process[57]. *p*-Pyrrd–Phen, the electron-injection efficiency of

which is comparable to that of PEI, is expected to be applied not only to OLEDs but also to OSCs since *p*-Pyrrd–Phen has advantages over PEI such as processability in a solution and vacuum, and applicability to CMETLs.

## Discussion

We conclude that the coordination reaction between Phen derivatives and electrodes is useful for producing a stable electrode with a low WF, and the change in the WF is mainly dominated by the amount of ET associated with the coordination reaction. In contrast to Phen-modified electrodes, the surface WF tuned by using conventional reactive materials, such as alkali metals and/or molecular n-dopants, is specific to the materials, and their air stability is compromised by the required low WF and IP. Overall, a Phen-modified electrode has many advantages over the conventional electron-injection materials, such as their WF tunability, ease of handling, high air and operational stability, and solution and vacuum processability. Since the Phen-modified electrode having the lowest WF exhibits better electron-injection efficiency than LiF, we believe that a great step has been made toward removing reactive materials from organic optoelectronic devices.

## Methods

**Ultraviolet photoelectron spectroscopy (UPS)**. Glass substrates coated with a 150-nm-thick ITO layer were cleaned with ultrapurified water, organic solvents and UV–ozone treatment. ZnO was deposited using a Mirror–Tron sputtering system (Choshu Industry Co., Ltd.). Then, we deposited 5-nm-thick Phen derivatives using a vacuum evaporation system, and samples were taken out into the air once. The samples were placed in a holder and then introduced into the load lock chamber of the UPS measurement apparatus. The total time of exposure to air was 10 min for all samples (Supplementary Fig. 8). UPS spectra of glass/ITO/ZnO/Phen derivatives were measured using a CHA analyser with a 128-channel detector; the excitation source was a HeI (21.22 eV) discharge lamp. A bias of −8.0 V was applied to each sample to separate the sample and the secondary edge for the analyser.

**DFT calculation**. Quantum chemical calculations were performed using the hybrid DFT functional Becke and Hartree–Fock exchange and the Lee Yang and Parr correlation (B3LYP) as implemented by the Gaussian 09 program packages. Electrons were described by Pople's 6–31G(d,p) basis sets for the five Phen derivatives (Supplementary Fig. 1). On the other hand, electrons were described by the LanL2DZ basis sets for the five Phen derivatives with Zn (Supplementary Fig. 3).

**Fabrication of inverted OLEDs and conventional OLEDs**. The inverted OLEDs shown in Fig. 2a were fabricated on glass substrates coated with a 150-nm-thick ITO layer. Prior to the fabrication of organic layers, the substrate was cleaned with ultrapurified water, organic solvents and UV–ozone treatment. ZnO was deposited using a Mirror–Tron sputtering system (Choshu Industry Co., Ltd.). The organic layers were sequentially deposited onto the substrate without breaking the vacuum at a pressure of approximately $10^{-5}$ Pa. The film structure of an inverted OLED is ITO/ZnO (3 nm)/Phen derivative (10 nm)/Zn(BTZ)$_2$ (10 nm)/Zn(BTZ)$_2$:Ir(piq)$_3$ (6 wt%, 15 nm)/α-NPD (40 nm)/HAT-CN (10 nm). DBTPB is N,N,N′-bis(dibenzo [b,d]thiophen-4-yl)-N4,N4′-diphenylbiphenyl-4,4′-diamine, α-NPD is 4,4′-bis[N-(1-naphthyl)-N-phenyl-amino]biphenyl and HAT-CN is 1,4,5,8,9,11-hexaaza-triphenylenehexacarbonitrile. After the organic layers were formed, a 100-nm-thick Al layer was deposited as the anode. The devices were encapsulated using a UV–epoxy resin and a glass cover in a nitrogen atmosphere after cathode formation.

In inverted OLED-2 shown in Fig. 3, the *p*-Pyrrd–Phen layer with a thickness of 10 nm was spin-coated in the atmosphere onto the ZnO surface from a cyclopentanone solution (2.5 g/L, 3000 rpm, 30 s). Then, the substrate was baked for 1 h at 150 °C to volatilise the solvent. In inverted OLED-3 shown in Fig. 3, PEI (P-1000, Nippon Shokubai Co., Ltd.) was spin-coated in the atmosphere onto the ZnO surface from an ethanol solution (0.5 wt%, 2000 rpm, 30 s) to give an ultrathin layer. The PEI layer was too thin to measure using a profilometer; however, its existence on the cathode was confirmed from the device characteristics. After spin casting, the substrate was annealed in ambient atmosphere (5 min, 150 °C). In inverted OLED-4 shown in Fig. 3, 1-nm-thick LiF was deposited onto the cleaned ITO surface. The other organic layers and the Al anode were sequentially deposited onto these substrates. The devices were encapsulated using a UV–epoxy resin and a glass cover in a nitrogen atmosphere after anode formation; however, the glass cover was removed at the time of the air stability test (Fig. 3d, e).

The conventional OLED shown in Fig. 4a was fabricated in a similar way to the above-mentioned OLEDs. After the UV–ozone treatment, Clevios HIL 1.3

(supplied by Heraeus Holding GmbH) was spun onto the substrate to form a 10-nm-thick layer. The other organic layers were sequentially deposited onto the substrate. The film structure of the conventional OLED is ITO (100 nm)/Clevios HIL 1.3 (10 nm)/α-NPD (20 nm)/4DBFP3Q (10 nm)/DIC-TRZ:Ir(mppy)$_3$ (3 wt%, 25 nm)/spB-BPy$_2$ (40 nm), where α-NPD is 4,4′-bis[N-(1-naphthyl)-N-phenyl-amino]biphenyl, 4DBFP3Q is N3,N3‴-bis(dibenzo[b,d]furan-4-yl)-N3,N3‴-diphenyl-[1,1′:2′,1″:2″,1‴-quaterphenyl]-3,3‴-diamine, Ir(mppy)$_3$ is *fac*-tris(3-methyl-2-phenylpyridinato-N,C2′-)iridium(III) and spB-BPy$_2$ is 6″-(4-([2,2′-bipyridin]-6-yl)-2-(5H-dibenzo[b,d]borolyl)phenyl)-2,2′:6′,3″-terpyridine. After the organic layers were formed, a 1-nm-thick EIL and a 100-nm-thick Al layer were deposited as the cathodes. The devices were encapsulated using a UV–epoxy resin and a glass cover in a nitrogen atmosphere after cathode formation.

**Device characterisation**. The electroluminescence (EL) spectra and luminance were measured using a spectroradiometer (Minolta CS-1000). A digital Source-Meter (Keithley 2400) and a desktop computer were used to operate the devices. We assumed that the emission from OLEDs was isotropic so that the luminance was Lambertian; thus, we calculated $\eta_{EQE}$ from the luminance, current density and EL spectra.

## Data availability
The data that support the plots within the paper are available from the corresponding author upon reasonable request.

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

## Acknowledgements

The authors thank Heraeus Holding GmbH for supplying Clevios HIL 1.3.

## Author contributions

H.F. supervised the project and wrote the paper. K.S., H.I., K.I. and T.S. conducted most of the experiments and analysed the data. M.H. and K.M. synthesised the materials. T.O. and T.S. analysed the data and revised the paper.

## Competing interests

The authors declare no competing interests.
