## [Peer Review File · Nature Communications]

Reviewers' Comments:

Reviewer #1:

Remarks to the Author:

The manuscript reported a series of Phen derivatives by utilising the coordination reaction to produce a stable electrode with various WFs, which can be tuned from 3.29–2.43 eV on the film of Phen derivative/ITO/ZnO. However, the molecular dipole moment, ESP and the amount of ET effects on ΔWF seem unconvincing for the paper's purpose. More work needs to be done to organize and sort out the relationship of gap state, ΔWF , Δe , driving voltage, and the electron injection/hole-blocking property. Therefore, I think there is still more work to do before publishing. The paper is inappropriate to be published in Nature Communications at present.

- 1) It's not accurate to contribute the ΔWF to molecular dipole moment, ESP and the amount of ET estimated by DFT calculation with a single molecule. The corresponding calculation should be performed on Zn coordination compounds.
- 2) Since the amount of ET showing the most potent effect on ΔWF , the amount of ET of the two-molecule system should be calculated for further confirm. Supported references would better be supplied at the position of "it was found that the calculated amount of ET of a two-molecule system". Besides, there's no logical connection between the ESP and ET.
- 3) Both 1H and ^{13}C NMR spectra and Mass spectrometric analysis (high resolution) should be afforded for new compounds. 1H NMR data of p-Pyrrd-Phen was missed in SI. Also, the identity and purity of the compounds should be demonstrated.
- 4) The wrong chemical structures of p-NMe₂-Phen and p-Pyrrd-Phen were presented in the part of "Supplementary Section 1: Materials".
- 5) ΔWF was shown in Figure 1, the detailed WF and specific evolution of work function and conductivity under photo-activation maybe introduced in the supporting information.
- 6) The mechanism of coordination reaction between p-NMe₂-Phen and Zn should be explained in detail.
- 7) Calculated IP using DFT calculation should be given in the supporting information and illustrated in the manuscript.
- 8) Compared with your previous work and other references, what is the advantages and disadvantages for this work? The list of OLED performance comparison may be listed.
- 9) There are many spoken transitional words, for example, "In addition" was used so many times. The author should consider changing the words.
- 10) Apparently, the idea of this work follows the reference "Nat. Commun. 10, 866 (2019)", and you have used the electrode ZnO rather than Ag to and tried some new phenanthroline derivatives. But my question is that, compared with the former work, what is the superiorities for your research.
- 11) In your MS, you have claimed the "coordination reaction" between ZnO and Phen derivatives, but there is not any experimental proof to support this conclusion, except for some calculations. But actually, the calculations should be as the supplementary or assistant proof to aid or further explain the experiments, rather than the crucial evidence.
- 12) The most serious problem is that you should more professionally reconsider and analyse your UPS data, and more importantly, make it clear the energy levels for charge injection and transport. In fact, it is little sense to evaluate the " ΔWF " which actually infers the shift of vacuum level. The most important information from UPS is the HOMO or gap state. Others, like what means for "two gap states" and "the original HOMO" in Figure 1d?
- 13) Please take care this, the thickness of 10 nm has beyond the range of solely electrode modification.
- 14) The right two figures of Figure 1f cannot give the clear information.
- 15) Apparently, the idea of this work follows the reference "Nat. Commun. 10, 866 (2019)", and you have used the electrode ZnO rather than Ag to and tried some new phenanthroline derivatives. But my question is that, compared with the former work, what is the superiorities for your research.
- 16) In your MS, you have claimed the "coordination reaction" between ZnO and Phen derivatives,

but there is not any experimental proof to support this conclusion, except for some calculations. But actually, the calculations should be as the supplementary or assistant proof to aid or further explain the experiments, rather than the crucial evidence.

17) The most serious problem is that you should more professionally reconsider and analyse your UPS data, and more importantly, make it clear the energy levels for charge injection and transport. In fact, it is little sense to evaluate the "ΔWF" which actually infers the shift of vacuum level. The most important information from UPS is the HOMO or gap state. Others, like what means for "two gap states" and "the original HOMO" in Figure 1d?

18) Please take care this, the thickness of 10 nm has beyond the range of solely electrode modification.

19) The right two figures of Figure 1f cannot give the clear information.

20) The EQE–luminance characteristics of p-NMe₂-Phen and p-Pyrrd-Phen were low than the existing Bphen under high current (Figure 2e). Why?

21) 22) "In addition, we see from Fig. 2e that the external quantum efficiency (EQE) of the inverted OLED using p-MeO-Phen is lower than that of the inverted OLED using BPhen, although p-MeO-Phen causes a larger ΔWF than BPhen. Thus, it is reasonable to assume that the characteristics of inverted OLEDs are dominated by not only ΔWF, but also the hole-blocking property around a cathode." How about the relationship between ΔWF and the hole-blocking property for the EQE of five Phen derivatives? Who plays the leading role?

22) In the evaluation of electron injection efficiency and air stability of cathode-modification layers in inverted OLEDs. The Luminance–voltage and current density–voltage characteristics of cOLED employing p-Pyrrd-Phen were almost the same as the PEI (Figure 3d-e). How about the advantages of p-Pyrrd-Phen than PEI?

23) Provide the details about the decay test in the light-emitting area of inverted OLEDs. It was clear that the initial brightness of inverted OLED-3 is significantly higher than others, but the Luminance–voltage and current density–voltage characteristics of OLED-2 and OLED-3 are almost the same (Figure 3b-d). The detailed measurement could directly affect the decay test.

24) "On the other hand, the relatively low operational stability of the cOLED employing p-MeO-Phen, where the electron injection efficiency is insufficient, may be caused by the accumulation of the carrier recombination region rather than the stability of p-MeO-Phen^{45–47}." The author explained the relatively low operational stability of the cOLED employing p-MeO-Phen was caused by the accumulation of the carrier recombination region rather than the stability of p-MeO-Phen. Please confirm it with detailed measurements and recheck the data.

25) Why the EQE–current density curves characteristic of cOLEDs employing p-MeO-Phen was higher than p-Pyrrd-Phen, but there was a significant difference in the Luminance–voltage and (inset) current density–voltage characteristics (Figure 4b-c)? How to explain? Please recheck the data.

Reviewer #2:

Remarks to the Author:

The authors demonstrates that low work function contacts can be achieved using phenanthroline-based interlayers that coordinate to the metal and ZnO cathodes. They further show for inverted OLEDs that the electron injection efficiency and air stability of a phenanthroline-modified electrode are higher than those of a LiF-modified electrode, the "benchmark" of OLEDs.

The authors carry out UPS experiments on in situ prepared cathodes and phenanthroline-based interlayers (5 nm) and show the formation of gap states that are attributed to through-bond charge transfer (nitrogen atoms bonding/coordinating with the Zn at the ZnO surface). The charge transfer at the interface is given as the main factor for the resulting low work function.

I find the paper to be well written and the results useful. I think the paper could be improved however with a few additions.

1. If I understood the paper correctly, the UPS experiments were carried on in situ prepared ZnO cathodes and phenanthroline-derivative films, whereas for the OLED 2, the ZnO were exposed to air before spin-coating the phenanthroline derivative (also in air). Here, the surface of the ZnO

may not be as reactive as for the in situ prepared ZnO/phenanthroline-derivative system and the WF-lowering mechanism may then be different (see e.g. J. Mater. Chem. C, 2020, 8, 173). Carrying out UPS of ZnO films exposed to air and then inserted into the UHV system for phenanthroline-derivative deposition may give a more precise comparison between the UPS and OLED 2.

2. The gap states evident in the UPS are convincing evidence for charge transfer given the WF shift. However, additional evidence of charge transfer from XPS or FTIR would be helpful, but this is not a "game breaker".

Reviewer #3:

Remarks to the Author:

The work by Fukagawa et al, presents a novel way to modify the workfunction of zinc oxide. The approach is based on the interaction between phenanthroline variaties and ZnO. Quite impressive changes in the WF are observed.

The authors present fundamental analysis of the modified electrodes and apply these electrodes in efficient OLEDs. The latter confirms that indeed the WF can be tuned as witnessed by the effect on the driving voltage.

I think this is a very nice work, and have only a few questions that would require some explanation or comment from the authors.

The authors use a very thin ZnO layer, what is the reason for such a thin layer. Is this required to obtain good electron injection? Are the authors certain the ITO is fully covered by such a thin ZnO layer? How is this verified?

Did the authors verify if these modified ZnO electrodes can also be effective charge extraction layers for photovoltaic devices?

What is the role of HAT-CN between the HTL and the metal electrode. Judging from the energy levels shown in Fig. 2. B, it would not be expected to lead to efficient doping of the HTL.

Our Responses to the Comments of the Reviewers

Reviewer #1

General comment: *The manuscript reported a series of Phen derivatives by utilising the coordination reaction to produce a stable electrode with various WFs, which can be tuned from 3.29–2.43 eV on the film of Phen derivative/ITO/ZnO. However, the molecular dipole moment, ESP and the amount of ET effects on ΔWF seem unconvincing for the paper's purpose. More work needs to be done to organize and sort out the relationship of gap state, ΔWF , Δe , driving voltage, and the electron injection/hole-blocking property. Therefore, I think there is still more work to do before publishing. The paper is inappropriate to be published in Nature Communications at present.*

Our response: We are grateful to the reviewer for recommending publication of the manuscript in Nature Communications after major revisions. Changes are shown by red-letter in the revised manuscript. (The Supplementary information is written in black letter.)

Comment 1: *It's not accurate to contribute the ΔWF to molecular dipole moment, ESP and the amount of ET estimated by DFT calculation with a single molecule. The corresponding calculation should be performed on Zn coordination compounds.*

Comment 2-1: *Since the amount of ET showing the most potent effect on ΔWF , the amount of ET of the two-molecule system should be calculated for further confirm. Supported references would better be supplied at the position of "it was found that the calculated amount of ET of a two-molecule system".*

Our response: Many thanks for these suggestions. To verify the validity of the calculation using single Zn atom, we compared the amount of ET from ethylamine to Zn with a periodic ZnO structure reported in Ref. 23 (Zhou *et al.*) with the amount of ET from ethylamine to metallic counterparts that we can treat in our calculations such as simple elements (Zn atom, ZnO). It was found that the Zn atom is more suitable than ZnO for the calculation of the ET from N in phenanthroline to the metal without periodic structure (Supplementary Fig. 4 is added). The following discussion has also been added in page 8 of the revised manuscript:

Before we discuss the correlation between the ET from Phen to the metal and Δ_{WF} , we must clarify the validity of using the amount of ET calculated by placing a Zn atom near N in Phen derivatives (Supplementary Fig. 2). We compared the amount of ET from ethylamine to Zn with a periodic ZnO structure reported by Zhou *et al.* with the amounts of ET from ethylamine to metallic counterparts that we can treat in our calculations such as simple elements (Zn atom, ZnO) (Supplementary Fig. 4)²³. We found that the Zn atom is more suitable than ZnO for the calculation of the ET from N to a metal without a periodic structure.

Comment 2-2: *Besides, there's no logical connection between the ESP and ET.*

Our response: Thank you for pointing that out. The manuscript has been changed as follows (page 8):

Original	Although the calculated ESP is an indicator of nucleophilicity, it was found that the calculated amount of ET of a two-molecule system could be a more accurate indicator of Δ_{WF} caused by a coordination reaction.
Corrected	It was found that the calculated amount of ET of a two-molecule system could be a good indicator of Δ_{WF} caused by a coordination reaction.

Comment 3: *Both 1H and ^{13}C NMR spectra and Mass spectrometric analysis (high resolution) should be afforded for new compounds. 1H NMR data of p-Pyrrd-Phen was missed in SI. Also, the identity and purity of the compounds should be demonstrated.*

Our response: Thank you for pointing that out. We have revised Supplementary Section 1 in accordance with the comment. Unfortunately, it is difficult to measure HR-Mass spectrum due to

the serious machine trouble. However, it is reasonable to suppose that phenanthroline derivatives shown in SI have successfully been synthesized if we consider the synthesis route and NMR spectra.

Comment 4: *The wrong chemical structures of p-NMe₂-Phen and p-Pyrrd-Phen were presented in the part of "Supplementary Section 1: Materials".*

Our response: We apologize for the wrong chemical structures of p-NMe₂-Phen and p-Pyrrd-Phen in Supplementary Section 1. We have revised their chemical structure.

Comment 5: *Δ WF was shown in Figure 1, the detailed WF and specific evolution of work function and conductivity under photo-activation maybe introduced in the Supplementary information.*

Our response: We apologize for the insufficient explanation about the UPS measurement. The shift in the cut-off position was not observed in the three repeated UPS measurements. Furthermore, we see from Fig. 4b and 4c (newly added data), where the correlation between electron injection efficiency of the Phen-modified Al electrode and Δ WF is illustrated, that the effect of photo-activation on the Δ WF is small. If Δ WF is affected by the emission of OLED, the luminance-current density-voltage characteristics are expected to show peculiar behaviour.

Comment 6: *The mechanism of coordination reaction between p-NMe₂-Phen and Zn should be explained in detail.*

Our response: We thank the reviewer for his/her constructive comment. As Yoshida reported (Ref. 26 in the manuscript), the correlation between the calculated ionization potential (IP) and the measured IP is useful information for discussing the coordination reaction. Thus, we organized data about the calculated IP and the measured IP in Supplementary Fig. 3, and the manuscript has been changed as follows (page 7):

Original	Such a gap state is also observed in other Phen derivatives on ITO/ZnO (Supplementary Fig. 4). From the UPS results, it is clear that the Phen derivative–Zn complexes illustrated in Fig. 1b are formed by the deposition of the Phen derivatives on ZnO (Supplementary Fig. 2).
Corrected	Such a gap state is also observed in other Phen derivatives on ITO/ZnO (Supplementary Fig. 4). Yoshida reported on complex formation between BCP and a Ag atom by comparing the electronic structure obtained by UPS with the calculated orbital energy ²⁶ . Here, the molecular orbitals and orbital energies were calculated by placing a Zn atom near N in Phen as in a previous report (Supplementary Fig. 2) ²⁶ . The highest occupied molecular orbital (HOMO) of the Phen derivative–Zn complex, which is the gap state, is localised on the Zn atom. On the other hand, the HOMO–1 of the Phen derivative–Zn complex mainly consists of the HOMO of each Phen derivative (Supplementary Figs. 1 and 2). For most Phen–Zn complexes, the IPs obtained by UPS are highly correlated with the calculated IPs, suggesting complex formation between Phen and the Zn atom (Supplementary Fig. 3).

Comment 7: *Calculated IP using DFT calculation should be given in the Supplementary information and illustrated in the manuscript.*

Our response: Many thanks for this suggestion. Summary of the calculated IP using DFT calculation is added in Supplementary Fig. 3.

Comment 8: *Compared with your previous work and other references, what is the advantages and disadvantages for this work? The list of OLED performance comparison may be listed.*

Our response: Many thanks for these suggestions. I suppose that there are some differences between this work and other references as follows. First, the number of Phen derivative used in this study is the maximum (five), whereas the number of metallic counterparts is one. However, the study on the coordination reaction between five Phen derivatives and one metallic counterpart leads a systematic understanding of the effect of coordination reaction on the Δ_{WF} for the first time. The following discussion has been added in pages 6~7 of the revised manuscript:

Although there have been many reports on the electronic structure and the related interactions between a specific Phen derivative and several metals, there have been few reports on the interactions between several Phen derivatives and a specific metal^{26, 37, 38}. Bin *et al.* were the first to use three Phen derivatives, and the *p*-MeO-Phen-Ag complex has been reported to be an excellent electron injection layer (EIL)³⁰. However, the three types of Phen used here do not provide a systematic understanding of the correlation between coordination reactions and Δ_{WF} around the cathode. We see from the calculated ESP that both *p*-NMe₂-Phen and *p*-Pyrrd-Phen have the potential to be better EILs than *p*-MeO-Phen when they are used around the cathode. The study of the interactions of five different Phen derivatives with metals will provide a systematic understanding of the effect of coordination reactions on Δ_{WF} .

Since the “*The list of OLED performance comparison*” is strongly correlated with **Comment 10**, we will respond to this comment below.

Comment 9: *There are many spoken transitional words, for example, “In addition” was used so many times. The author should consider changing the words.*

Our response: Thank you for pointing that out. Some “In addition” are changed to other words or deleted.

Comment 10: *Apparently, the idea of this work follows the reference “Nat. Commun. 10, 866 (2019)”, and you have used the electrode ZnO rather than Ag to and tried some new phenanthroline derivatives. But my question is that, compared with the former work, what is the superiorities for your research.*

Our response: Many thanks for this suggestion. One superiority, a systematic understanding of the effect of coordination reaction on the Δ_{WF} , is provided as a response to **Comment 8** (see above). We would like to emphasise that the following discussion represents the most important superiority for our research.

The study of the interactions of five different Phen derivatives with metals will provide a systematic understanding of the effect of coordination reactions on Δ_{WF} .

Secondly, the superiority of using ZnO is written in the manuscript as follows (pages 5~6).

the changes in the surface WF observed by the deposition of Phen can be discussed as the effect of the coordination reaction, since there is no push-back effect on metal-oxide surfaces unlike metal surfaces³⁵.

Thirdly, the electron injection efficiency of the *p*-Pyrrd-Phen (this work)-modified Al is demonstrated to be higher than that of *p*-MeO-Phen (previous work)-modified Al. The manuscript (page 13) has been changed as follows:

Original	Although p -MeO-Phen is reported to have a strong nucleophilicity ³⁰ , it is difficult to inject electrons from the p -MeO-Phen-modified Al.
Corrected	The electron injection efficiency of p -Pyrrd-Phen-modified Al is demonstrated to be higher than that of p -MeO-Phen-modified Al ³⁰ .

Fourthly, there is a difference in applicability of the electron injection layer (EIL) to the OLED production line. In the previous work, Ag-doping is proposed to essential to ensure high electron injection efficiency. However, the organic layer deposition and metal deposition chambers are

generally separated in the OLED production line. We summarised “*the list of OLED performance comparison*” with the applicability of EILs to the production line in Supplementary Table. 1, and the following discussion has been added in pages 13~14 of the revised manuscript

Although Ag-doped *p*-MeO-Phen may exhibit similar electron injection efficiency to *p*-Pyrrd-Phen-modified Al, it is not easy to apply such a Ag-doped film to an OLED production line, where the organic layer deposition and metal deposition chambers are generally separated (Supplementary Table 1)⁵⁴.

Comment 11: *In your MS, you have claimed the “coordination reaction” between ZnO and Phen derivatives, but there is not any experimental proof to support this conclusion, except for some calculations. But actually, the calculations should be as the supplementary or assistant proof to aid or further explain the experiments, rather than the crucial evidence.*

Our response: We apologize for the insufficient explanation about coordination reaction and related reference. This comment is strongly correlated with **Comment 6**. Furthermore, the chemical interaction between *p*-Pyrrd-Phen and Zn was confirmed by using X-ray photoelectron spectroscopy (Supplementary Fig. 6). The following discussion has been added in page 7 of the revised manuscript:

Yoshida reported on complex formation between BCP and a Ag atom by comparing the electronic structure obtained by UPS with the calculated orbital energy²⁶. Here, the molecular orbitals and orbital energies were calculated by placing a Zn atom near N in Phen as in a previous report (Supplementary Fig. 2)²⁶. The highest occupied molecular orbital (HOMO) of the Phen derivative–Zn complex, which is the gap state, is localised on the Zn atom. On the other hand, the HOMO–1 of the Phen derivative–Zn complex mainly consists of the HOMO of each Phen derivative (Supplementary Figs. 1 and 2). For most Phen–Zn complexes, the IPs obtained by UPS are highly correlated with the calculated IPs, suggesting complex formation between Phen and the Zn atom (Supplementary Fig. 3). Furthermore, the chemical interaction between *p*-Pyrrd-Phen and Zn was confirmed by X-ray photoelectron spectroscopy (Supplementary Fig. 6).

Comment 12-1: *The most serious problem is that you should more professionally reconsider and analyse your UPS data, and more importantly, make it clear the energy levels for charge injection and transport. In fact, it is little sense to evaluate the “ ΔWF ” which actually infers the shift of vacuum level. The most important information from UPS is the HOMO or gap state.*

Our response: We apologize for the lack of explanation of the correlation between the UPS data and the purpose of this study. The main purpose of this study is producing a stable electrode with various WFs by utilising the coordination reaction. Behind this background, the method of improving the electron injection/collection efficiency without using reactive materials has been desired as written in the introduction. Thus, the information on ΔWF is more important than the information of the valence band or the gap state. On the other hand, the information on the gap state is quite important for discussing the coordination reaction as we responded above (**Comment 6** and **Comment 11**). To emphasize the importance of ΔWF , the following discussion has been added in page 10 of the revised manuscript:

The correlation between the value of ΔWF derived from the Phen derivatives and the electron injection efficiency, which is of great significance toward the realisation of organic devices without using reactive metals, was successfully clarified by using Phen derivatives for the EIL of conventional OLEDs (cOLEDs) as will be shown in Fig. 4.

Comment 12-2: *Others, like what means for “two gap states” and “the original HOMO” in Figure 1d?*

Our response: Thank you for pointing that out. After reconsidering the shape of the gap state with reference to the previous reports (N. Ueno et al. Progress in Surface Science, vol.83, pp. 490-557 and S. Kera et al. Progress in Surface Science, vol.84, pp. 135-154), we decided that it would not be

appropriate to conclude the number of gap states to two. Thus, we changed it to “the gap state”. The manuscript has been changed as follows (page 7):

Original	In the UPS spectrum of BPhen on ITO/ZnO, two gap states are observed between the Fermi level (E_F) and the original highest occupied molecular orbital as in the case with BCP–metal complex systems ^{26,37} .
Corrected	In the UPS spectrum of BPhen on ITO/ZnO, the gap state is observed near the Fermi level (E_F) as in the case with BCP–metal complex systems ^{26,37} .

In addition, definition of gap state (HOMO) or original HOMO has been changed as follows (page 7):

The highest occupied molecular orbital (HOMO) of the Phen derivative–Zn complex, which is the gap state, is localised on the Zn atom. On the other hand, the HOMO–1 of the Phen derivative–Zn complex mainly consists of the HOMO of each Phen derivative (Supplementary Figs. 1 and 2).

Comment 13: *Please take care this, the thickness of 10 nm has beyond the range of solely electrode modification.*

Our response: Many thanks for this suggestion. We redefined this 10-nm-thick layer to be cathode-modification and electron transport layer (CMETL).

Comment 14: *The right two figures of Figure 1f cannot give the clear information.*

Our response: Thank you for pointing that out. We have added names of material to each plot.

The original Comment 15 ~ 19 are the same as original Comment 10 ~ 14

Comment 15 (20): *The EQE–luminance characteristics of p-NMe2-Phen and p-Pyrrd-Phen were low than the existing Bphen under high current (Figure 2e). Why?*

Comment 16 (21, 22): *“In addition, we see from Fig. 2e that the external quantum efficiency (EQE) of the inverted OLED using p-MeO-Phen is lower than that of the inverted OLED using BPhen, although p-MeO-Phen causes a larger ΔW_F than BPhen. Thus, it is reasonable to assume that the characteristics of inverted OLEDs are dominated by not only ΔW_F , but also the hole-blocking property around a cathode.” How about the relationship between ΔW_F and the hole-blocking property for the EQE of five Phen derivatives? Who plays the leading role?*

Our response: Thank you for pointing that out. Since the holes that reach around the Phen derivative make the interpretations of the characteristics of inverted OLEDs complicated, it is difficult to simply discuss the correlation between ΔW_F and the electron injection efficiency in inverted OLEDs with ITO/ZnO as the cathode. On the other hand, the correlation between the ΔW_F and the electron injection efficiency was successfully clarified by using Phen derivatives for the EIL of conventional OLEDs (cOLEDs), which is added in Fig. 4. The manuscript has been changed as follows:

(pages 9~10)

Original	Thus, it is reasonable to assume that the characteristics of inverted OLEDs are dominated by not only ΔW_F , but also the hole-blocking property around a cathode, as Bolink et al. reported ³² . Actually, the hole-blocking property of the BPhen–Zn complex is expected to be superior to that of the p-MeO-Phen–Zn complex since there is a clear difference in energy of the HOMO, as shown in Fig. 1d. Furthermore, we see from the difference in the energy of the gap state, as shown in Fig. 1d, that the hole-blocking property of the p-Pyrrd-Phen–Zn complex is expected to be superior to that of the p-NMe2-Phen–Zn complex, resulting in the higher EQE in the inverted OLED using p-Pyrrd-Phen. The fact that the p-Pyrrd-Phen–Zn complex combines good
----------	---

	electron injection and hole-blocking properties suggests that p -Pyrrd-Phen is the most promising material for producing a Phen-modified electrode with a low WF.
Corrected	Thus, it is reasonable to assume that the characteristics of inverted OLEDs are dominated by not only Δ_{WF} , but also the hole-blocking property around a cathode, as Bolink et al. reported ³² . This is because the holes are effectively injected or generated at 4,4'-bis[N-(1-naphthyl)-N-phenyl-amino]biphenyl (α -NPD)/1,4,5,8,9,11-hexaazatriphenylenehexacarbonitrile (HAT-CN)/Al interfaces ⁴⁰ . The highest EQE of the inverted OLED fabricated using BPhen at a high current density is expected to originate from the strongest hole-blocking property at the BPhen/emitting layer interface. It is reasonable to suppose that the hole-blocking property at the Phen derivative/emitting layer interface is determined by the orbital energy of the HOMO of each Phen derivative (not the complex with Zn), whereas the hole-blocking property at the Phen derivative/ZnO interface is determined by the orbital energy of the gap state. We see from the difference in the energy of the gap state, as shown in Fig. 1d, that the hole-blocking property of the p -Pyrrd-Phen-Zn complex around the cathode is expected to be superior to that of the p -NMe ₂ -Phen-Zn complex, resulting in the higher EQE in the inverted OLED using p -Pyrrd-Phen. The fact that the p -Pyrrd-Phen-Zn complex combines good electron injection and hole-blocking properties suggests that p -Pyrrd-Phen is the most promising material for producing a Phen-modified electrode with a low WF. Since the holes that reach the Phen derivative make the interpretation of the characteristics of inverted OLEDs complicated, it is difficult to simply discuss the correlation between Δ_{WF} and the electron injection efficiency in inverted OLEDs with ITO/ZnO as the cathode, which was used for the evaluation of Δ_{WF} in Fig. 1. However, the characteristics of inverted OLEDs shown in Fig. 2 greatly contributed to the evaluation of the air stability of the electrode modified by Phen derivatives, as will be shown in Fig. 3. The correlation between the value of Δ_{WF} derived from the Phen derivatives and the electron injection efficiency, which is of great significance toward the realisation of organic devices without using reactive metals, was successfully clarified by using Phen derivatives for the EIL of conventional OLEDs (cOLEDs) as will be shown in Fig. 4.

(page 12)

Original	Finally, we show that a Phen derivative can be a feasible alternative to popular but reactive EILs such as LiF and 8-Quinolinolato lithium (Liq) ⁴¹ . To this end, we evaluated the EIL-dependent characteristics of a conventional OLED (cOLED) having the device configuration depicted in Fig. 4a, employing various EILs ⁴² . Since Phen derivatives can be deposited differently from PEI, it is possible to evaluate the electron injection efficiency in cOLEDs. Although some evaporated Al atoms diffuse into Phen derivatives in cOLEDs ⁴³ , we can conjecture that the electron injection efficiency of the Phen-modified Al electrode basically corresponds to Δ_{WF} shown in Fig. 1d. We see from Fig. 4b that
Corrected	The correlation between the value of Δ_{WF} derived from the Phen derivatives and the electron injection efficiency has successfully been observed by evaluating the EIL-dependent characteristics of a cOLED having the device configuration depicted in Fig. 4a, employing various EILs ⁴⁵ . The emitting host we selected here is a thermally activated delayed fluorescent (TADF) material named 2,4-diphenyl-6-bis(12-phenylindolo)[2,3-a]carbazol-11-yl)-1,3,5-triazine (DIC-TRZ), which is ideal for demonstrating an efficient and operationally stable OLED ⁴⁶ . Although some evaporated Al atoms diffuse into Phen derivatives in cOLEDs ⁴⁷ , we see from Figs. 4b and 4c that the electron injection efficiency of the Phen-modified Al electrode basically corresponds to Δ_{WF} shown in Fig. 1d (Supplementary Fig. 7). The result of our experiment clearly shows that the electron injection efficiency can easily be controlled by changing the Phen derivative around the cathode. Furthermore, we show that a Phen derivative can be a feasible alternative to popular but reactive EILs such as LiF and 8-Quinolinolato lithium (Liq) ⁴⁸ . To discuss the difference between this study and the previous study in detail, the characteristics of cOLEDs employing p -Pyrrd-Phen, p -OMe-Phen, LiF and Liq are summarised in Figs. 4d, 4e and 4f. We see from Fig. 4d that

Comment 17 (22): In the evaluation of electron injection efficiency and air stability of cathode-modification layers in inverted OLEDs. The Luminance–voltage and current density–voltage characteristics of cOLED employing *p*-Pyrrd-Phen were almost the same as the PEI (Figure 3d-e). How about the advantages of *p*-Pyrrd-Phen than PEI?

Our response: Many thanks for this suggestion. The following discussion has been added in page 14 of the revised manuscript:

We would also like to mention an advantage of Phen derivatives with a high electron injection efficiency such as *p*-Pyrrd-Phen over PEI, which is widely used as an EIL in inverted OLEDs^{57,58}. Amine-based polymers such as PEI have also been used in OSCs as a cathode modification layer instead of BCP and/or BPhen owing to their high WF tunability⁵⁹. However, there are two major problems for the practical application of PEI since PEI can only be prepared by a solution process. One is that it is not easy to prepare multilayer structures by a solution process when PEI is applied to devices with a top cathode such as cOLEDs⁶⁰. The other is that the characteristics of devices fabricated using PEI are rather sensitive to the thickness of the PEI film, even though it is difficult to precisely control the film thickness by a solution process⁵⁷. *p*-Pyrrd-Phen, the electron injection efficiency of which is comparable to that of PEI, is expected to be applied not only to OLEDs but also to OSCs since *p*-Pyrrd-Phen has advantages over PEI such as processability in a solution and vacuum, and applicability to CMETLs.

Comment 18 (23): *Provide the details about the decay test in the light-emitting area of inverted OLEDs. It was clear that the initial brightness of inverted OLED-3 is significantly higher than others, but the Luminance–voltage and current density–voltage characteristics of OLED-2 and OLED-3 are almost the same (Figure 3b-d). The detailed measurement could directly affect the decay test.*

Our response: We apologize for the lack of explanation of the decay test in the light emitting area. The following discussion has been added in the figure caption (Fig. 3d):

Although dc current was applied to the inverted OLEDs at the time of measurement, dc current was not applied under the storage condition.

Thus, the effect of initial brightness on the decay in light-emitting area is small.

Comment 19 (24): *“On the other hand, the relatively low operational stability of the cOLED employing *p*-MeO-Phen, where the electron injection efficiency is insufficient, may be caused by the accumulation of the carrier recombination region rather than the stability of *p*-MeO-Phen^{45–47}.” The author explained the relatively low operational stability of the cOLED employing *p*-MeO-Phen was caused by the accumulation of the carrier recombination region rather than the stability of *p*-MeO-Phen. Please confirm it with detailed measurements and recheck the data.*

Our response: Thank you for pointing that out. Very recently, our paper, where the accumulation of the carrier recombination region significantly shortens the operational lifetime of OLED, is published (Fukagawa et al. Adv. Opt. Mater., Ref. 53). In this recently-published paper, we report that differences in the charge injection barrier have a significant impact on the operational lifetime of OLEDs. Thus, we have added Ref. 53 in the revised manuscript.

Comment 20 (25): *Why the EQE–current density curves characteristic of cOLEDs employing *p*-MeO-Phen was higher than *p*-Pyrrd-Phen, but there was a significant difference in the Luminance–voltage and (inset) current density–voltage characteristics (Figure 4b-c)? How to explain? Please recheck the data.*

Our response: Thank you for pointing that out. The host material used in this study, which is a thermally activated delayed fluorescent (TADF) material named 2,4-diphenyl-6-bis(12-phenylindolo)[2,3-a] carbazol-11-yl)-1,3,5-triazine (DIC-TRZ), plays a key role for the observed driving voltage-independent EQE. The efficient energy transfer is possible from TADF host to the phosphorescent dopant (newly added Refs. 46 and 56), and the similar driving voltage-independent EQE as Fig. 4 had been reported for red phosphorescent OLEDs (newly added Ref. 55). The following discussion has been added in page 12 and 14 of the revised manuscript:

(page 12)

The emitting host we selected here is a thermally activated delayed fluorescent (TADF) material named 2,4-diphenyl-6-bis(12-phenylindolo)[2,3-a] carbazol-11-yl)-1,3,5-triazine (DIC-TRZ), which is ideal for demonstrating an efficient and operationally stable OLED⁴⁶.

(page 14)

It is no wonder that the EQEs of the OLEDs employing Phen derivatives are almost independent of the driving voltage since the EQEs independent of the driving voltage have also been reported for another phosphorescent OLEDs fabricated using a TADF material as the host^{55,56}.

Reviewer #2

General comment: The authors demonstrate that low work function contacts can be achieved using phenanthroline-based interlayers that coordinate to the metal and ZnO cathodes. They further show for inverted OLEDs that the electron injection efficiency and air stability of a phenanthroline-modified electrode are higher than those of a LiF-modified electrode, the "benchmark" of OLEDs.

The authors carry out UPS experiments on *in situ* prepared cathodes and phenanthroline-based interlayers (5 nm) and show the formation of gap states that are attributed to through-bond charge transfer (nitrogen atoms bonding/coordinating with the Zn at the ZnO surface). The charge transfer at the interface is given as the main factor for the resulting low work function.

I find the paper to be well written and the results useful. I think the paper could be improved however with a few additions.

Our response: We are grateful to the reviewer for recommending publication of the manuscript in Advanced Materials after some revisions. Changes are shown by red-letter in the revised manuscript. (The Supplementary information is written in black letter.)

Comment 1: If I understood the paper correctly, the UPS experiments were carried on *in situ* prepared ZnO cathodes and phenanthroline-derivative films, whereas for the OLED 2, the ZnO were exposed to air before spin-coating the phenanthroline derivative (also in air). Here, the surface of the ZnO may not be as reactive as for the *in situ* prepared ZnO/phenanthroline-derivative system and the WF-lowering mechanism may then be different (see e.g. *J. Mater. Chem. C*, 2020, 8, 173). Carrying out UPS of ZnO films exposed to air and then inserted into the UHV system for phenanthroline-derivative deposition may give a more precise comparison between the UPS and OLED 2.

Our response: We apologize for the insufficient explanation about the UPS measurement. Unfortunately, we do not have *in situ* UPS measurement system. It is inevitable that the sample (Phen derivatives on ITO/ZnO) is exposed to the air for about 10 minutes before UPS measurement. Thus, we evaluated the work function (WF) of *p*-Pyrrd-Phen on ITO/ZnO for different air exposure times (Supplemental Fig. 9). As in the case with previous report (Zhou et al. Supplemental Ref. 1), the effect of air exposure on WF is small especially for the short exposure time. The manuscript has been changed as follows (page 22, Methods):

Original	Then, we deposited 5-nm-thick Phen derivatives. UPS spectra of
Corrected	Then, we deposited 5-nm-thick Phen derivatives using a vacuum evaporation system, and samples were taken out into the air once. The samples were placed in a holder and then introduced into the load lock chamber of the UPS measurement apparatus. The total time of exposure to air was 10 min for all samples (Supplementary Fig. 9). UPS spectra of

As you can see from Supplementary Fig. 9, the effect of the short air exposure on Δ_{WF} is small, and the direction of Δ_{WF} is opposite to the difference in the characteristics of inverted OLED-1 and inverted OLED-2. Thus, it is reasonable to suppose that interpenetrated ZnO in inverted OLED-2 may contribute to electron injection/transport, as written in the manuscript.

In addition, many thanks for the information (*J. Mater. Chem. C*, 2020, 8, 173). The paper is added as Ref. 38.

Comment 2: The gap states evident in the UPS are convincing evidence for charge transfer given the WF shift. However, additional evidence of charge transfer from XPS or FTIR would be helpful, but this is not a "game breaker".

Our response: Many thanks for this suggestion. The chemical interaction between *p*-MeO-Phen and Zn was confirmed by using X-ray photoelectron spectroscopy (Supplementary Fig. 6). The following discussion has been added in page 7 of the revised manuscript:

Furthermore, the chemical interaction between *p*-Pyrrd-Phen and Zn was confirmed by X-ray photoelectron spectroscopy (Supplementary Fig. 6).

Reviewer #3

General comment: *The work by Fukagawa et al, presents a novel way to modify the workfunction of zinc oxide. The approach is based on the interaction between phenanthroline variaties and ZnO. Quite impressive changes in the WF are observed.*

The authors present fundamental analysis of the modified electrodes and apply these electrodes in efficient OLEDs. The latter confirms that indeed the WF can be tuned as witnessed by the effect on the driving voltage.

I think this is a very nice work, and have only a few questions that would require some explanation or comment from the authors.

Our response: We are grateful to the reviewer for recommending publication of the manuscript in Nature Communications after some revisions. Changes are shown by red-letter in the revised manuscript. (The Supplementary information is written in black letter.)

Comment 1: *The authors use a very thin ZnO layer, what is the reason for such a thin layer. Is this required to obtain good electron injection? Are the authors certain the ITO is fully covered by such a thin ZnO layer? How is this verified?*

Our response: We apologize for the insufficient explanation about employing ZnO. As the reviewer #3 pointed out, ZnO is essential to obtain good electron injection since ZnO is effective to lower the surface WF (as shown in Refs. 5 and 32). Although it is difficult to verify the coverage of thin ZnO layer, the surface WF is determined by the average of the point charges on the surface (added Ref. 33). Thus, ZnO is effective for lowering the surface WF independent of the growth behaviour except in the case of large-island growth under special growth conditions (added Ref. 34). The manuscript has been changed as follows (page 5):

Original	Secondly, the WF of ZnO is smaller than that of ITO; thus, ITO/ZnO is more favourable both for producing a low WF. Lastly,
Corrected	Secondly, the WF of ZnO is smaller than that of ITO; thus, ITO/ZnO is more favourable for both producing a low WF and enhancing the electron injection from ITO ^{5,32} . Since the surface WF is determined by the average of the point charges on the surface, it is reasonable to suppose that the reductions of both the surface WF observed by ultraviolet photoemission spectroscopy (UPS) measurements and the electron injection barrier are independent of the growth behaviour of ZnO except in the case of large-island growth under special growth conditions ^{33,34} . Lastly,

Comment 2: *Did the authors verify if these modified ZnO electrodes can also be effective charge extraction layers for photovoltaic devices?*

Our response: Thank you for pointing that out. Unfortunately, it is difficult for us to verify the applicability of Phen-modified ZnO electrodes to photovoltaic devices due to the lack of measurement system. However, it is reasonable to suppose that Phen-modified ZnO electrodes is applicable to photovoltaic devices. In the newly added Ref. 58, interfacial materials for organic solar cells are summarised. The interfacial material around cathode has been changed from BCP/BPhen to other materials that involve high electron injection/extraction efficiency such as polyethyleneimine (PEI). Since the electron injection efficiency of *p*-Pyrrd-Phen is demonstrated to be comparable to that of PEI, *p*-Pyrrd-Phen is also expected for the cathode modification layer. In addition, there are other advantages of employing *p*-Pyrrd-Phen over PEI such as processability in solution and vacuum, and the electron transportability (applicability to cathode modification and electron-transporting layer). The following discussion has also been added in page 14 of the revised manuscript:

Amine-based polymers such as PEI have also been used in OSCs as a cathode modification layer instead of BCP and/or BPhen owing to their high WF tunability⁵⁹. However, there are two major problems for the practical application of PEI since PEI can only be prepared by a solution process. One is that it is not easy to prepare multilayer structures by a solution process when PEI is applied to devices with a top cathode such as COLEDs⁶⁰. The other is that the characteristics of devices fabricated using PEI are rather sensitive to the

thickness of the PEI film, even though it is difficult to precisely control the film thickness by a solution process⁵⁷. *p*-Pyrrd-Phen, the electron injection efficiency of which is comparable to that of PEI, is expected to be applied not only to OLEDs but also to OSCs since *p*-Pyrrd-Phen has advantages over PEI such as processability in a solution and vacuum, and applicability to CMETLs.

Comment 3: *What is the role of HAT-CN between the HTL and the metal electrode. Judging from the energy levels shown in Fig. 2. B, it would not be expected to lead to efficient doping of the HTL.*

Our response: Many thanks for this suggestion. The following discussion has been added in page 9 of the revised manuscript:

This is because the holes are effectively injected or generated at 4,4'-bis[N-(1-naphthyl)-N-phenyl-amino]biphenyl (α -NPD)/1,4,5,8,9,11-hexaazatriphenylenehexacarbonitrile (HAT-CN)/Al interfaces⁴⁰.

Reviewers' Comments:

Reviewer #1:

Remarks to the Author:

The revision is good to me. it can be accepted now.

Reviewer #2:

Remarks to the Author:

I am satisfied by the authors' modifications to the manuscript and recommend its publication.

Reviewer #3:

Remarks to the Author:

The authors have adequately addressed my initial concerns.

I think the manuscript is now ready for publication.